# Optimized Planning and Evaluation of Dental Implant Fatigue Testing: A Specific Software Application

**DOI:** 10.3390/biology9110372

**Published:** 2020-10-31

**Authors:** Marta García-González, Sergio Blasón-González, Ismael García-García, María Jesús Lamela-Rey, Alfonso Fernández-Canteli, Ángel Álvarez-Arenal

**Affiliations:** 1Department of Prosthodontics and Occlusion, School of Dentistry, University of Oviedo, C/. Catedratico Serrano s/n., 33006 Oviedo, Spain; arenal@uniovi.es; 2Department of Component Safety, Bundesanstalt für Materialforschung und –Prüfung (BAM), Unter den Eichen 87, Germany BAM, 12205 Berlin, Germany; sergioblasonglez@gmail.com; 3Department of Construction and Manufacturing Engineering, University of Oviedo, Campus de Viesques, 33203 Gijón, Spain; UO195825@uniovi.es (I.G.-G.); mjesuslr@uniovi.es (M.J.L.-R.); afc@uniovi.es (A.F.-C.)

**Keywords:** dental materials, prostheses and implants, reference standards, software, cyclic loading, fatigue, lifetime, S-N curve, staircase method

## Abstract

**Simple Summary:**

Rehabilitation of missing teeth using dental implants is a common treatment option nowadays because of its high predictability. However, it is not exempt from complications such as mechanical failures, which are often related to implant and prosthesis design. Therefore, they must be tested before they are released to the market. The current regulations on in vitro fatigue tests, which ensure reliability in the mechanical function of dental implants, may be difficult to enforce due to their long duration and the high number of tests required. This work assesses this problem and proposes the use of the free software ProFatigue to optimize the process. Based on a well-known fatigue model, this software enables the researcher to determine more accurately the fatigue limit of the evaluated dental implant, as well as to reduce the testing costs due to a more suitable planning strategy of the in vitro test program. Consequently, this software may help researchers to improve the efficiency of this kind of mechanical test, which will eventually benefit those patients who may need implant-supported dental prostheses.

**Abstract:**

Mechanical complications in implant-supported fixed dental prostheses are often related to implant and prosthetic design. Although the current ISO 14801 provides a framework for the evaluation of dental implant mechanical reliability, strict adherence to it may be difficult to achieve due to the large number of test specimens which it requires as well as the fact that it does not offer any probabilistic reference for determining the endurance limit. In order to address these issues, a new software program called ProFatigue is presented as a potentially powerful tool to optimize fatigue testing of implant-supported prostheses. The present work provides a brief description of some concepts such as load, fatigue and stress-number of cycles to failure curves (S-N curves), before subsequently describing the current regulatory situation. After analyzing the two most recent versions of the ISO recommendation (from 2008 and 2016), some limitations inherent to the experimental methods which they propose are highlighted. Finally, the main advantages and instructions for the correct implementation of the ProFatigue free software are given. This software will contribute to improving the performance of fatigue testing in a more accurate and optimized way, helping researchers to gain a better understanding of the behavior of dental implants in this type of mechanical test.

## 1. Introduction

The high survival rates of implants, implant-supported single crowns and fixed dental prostheses over 97–98% at 5 years [1] make implant-supported rehabilitation a reliable treatment to replace missing teeth. However, it is not exempt from complications, which may be mechanical/technical, biological or esthetic [2,3]. The main effort of clinicians should be aimed at minimizing or avoiding all those factors that could eventually lead to failure, including component fracture or progressive bone loss. While there are a wide variety of contributing factors, they can be generally categorized as either patient-dependent (hygiene, peri-implant microbiota and bone quality) or professional-dependent (implant selection, restoration design, occlusal adjustment and other factors related to a possible load increment in the implant–restoration complex). The end result of these biomechanical factors is the transmission of stress to the prostheses, prosthetic components (like screw and abutments), implants and peri-implant bone. If these factors are not considered carefully enough, the resulting stress value can reach the implant material’s breaking strain, leading to its fracture, or the level of pathological overload, which may favor bone loss if the resorption process predominates over the bone formation [4,5].

Although controlled clinical tests with random allocation are the best method to evaluate the influence of these biomechanical factors, there are a number of ethical, deontological and economic issues that can hinder their design. However, the use of in vitro test methods allows for the recreation of any biomechanical condition in order to assess its effect on implants, peri-implant bone and implant–prothesis complex.

Numerical simulation analysis by finite elements and its validation through extensometry, photoelasticity, photogrammetry, fractal analysis and, most recently, DIC (digital image correlation) [6,7,8,9,10] are some of the most commonly used in vitro methods to evaluate results according to the load conditions and other characteristics of a given experiment. Anyway, since the simulation demands experimental verification to determine the long-term resistance of the components under real loads, it is of vital importance to conduct fatigue tests under the most realistic loads and effective chewing conditions possible.

## 2. General Considerations: Load, Fatigue, S-N Curves

Implant-supported prostheses receive loads from chewing in a wet environment and are subjected to abrupt changes in temperature, ranging from 0 to 60 °C. This means that the materials used must satisfy certain conditions in terms of hardness and stiffness, as well as resistance against deformation, impact fracture, fatigue fracture, dissolution, absorption, pigmentation, corrosion and low conductivity and thermal expansion. Moreover, all of these factors must be incorporated effectively into an appropriate design [11].

In vitro research on material properties and different crown–abutment–implant system behavior allows for the determination of the minimum requirements necessary for satisfactory intraoral function. These studies are essential to verify whether or not the new materials or designs provide actual improvements compared to those already on the market, before they are used in patients, so as to enable clinicians to choose the most appropriate system for each particular patient.

Mechanical testing is one of the physical tests that studies material behavior against loads. The implant–restoration complex may withstand different kinds of forces or loads (tensile, compressive, tangential, bending or torsion forces), so it is necessary to evaluate the resistance of the materials involved in order to determine the maximum load that they can bear before fracturing. Moreover, loads may fluctuate over time during chewing, reaching levels that are below the fracture load but which can nonetheless generate weaknesses in the structure along with loss of stiffness as a result of the appearance of fissures and cracks. This phenomenon is known as mechanical fatigue. Such small defects and cracks can gradually develop and propagate, leading to damage accumulation and eventually a fatigue fracture. The fatigue limit corresponds to the maximum load level that a material can withstand without fracture for a theoretically infinite number of load cycles [12,13].

Fatigue studies can be addressed through stress-based models, strain-based models and fracture mechanics-based models. The first approach, most common in dental research, uses S-N curves and the staircase method as experimental techniques to obtain basic information related to fatigue behavior and therefore to predict the overall lifetime.

The S-N curve (stress-number of cycles) or Wöhler curve (see Figure 1) is the graphic representation of the number of cycles that a specimen can endure before fracture when it is subjected to a different range of cyclic loads. This range is established by the difference between an upper and lower requested limit, but it remains constant during the testing [14]. Thus, the ordinate axis represents the force range expressed in Newtons (in a lineal or logarithmic scale) or, more commonly, the stress range expressed in N/mm^2^. The abscissa axis shows the applied number of cycles (always in a logarithmic scale). These graphics demonstrate that higher loads result in fracture at a relatively low number of cycles, while lower load levels request a larger number of cycles before failure occurs. The advantage of this model is that it allows for the S-N field to be interpreted as the representation of an accelerated life testing, enabling us to capture the component response in lifetime terms (number of cycles applied under a determined frequency) to multiple load levels. It also provides evidence of a fatigue limit existence. However, depending on the intended model, it is often necessary to test a relatively high number of specimens at different load levels, which, from an economic and time-saving perspective, is the opposite of an approach based on limited available results. In general, the evaluation of fatigue results may be carried out under probabilistic criteria, giving rise to the so-called S-N-P field.

In order to establish a reference load in the initial experiments, it is recommendable to first perform a static fatigue test, which applies an increasing monotonic load until failure occurs. To conduct the experimental program, appropriate equipment (such as a universal testing machine) must be used [10]. Similarly, given that the time needed for the specimens to fail is *a priori* unknown, it must be anticipated that the tests may last longer than expected. For this reason, researchers usually set a determined number of cycles (generally, 2, 5 or 10 million), which is mistakenly identified as the material fatigue limit for an infinite number of cycles [9]. Assuming that an individual eats three 15-min meals per day and that each chewing cycle is completed at a frequency of 1 Hz, approximately 2700 mastications would be performed per day. Thus, an estimated one million cycles of chewing sequences would be equivalent to one year of masticatory function [15]. In dentistry, several studies use 5 million cycles (around 5 years of masticatory function) as a reference to obtain the fatigue limit [16,17,18,19,20]. Given the complex nature of the masticatory function, such a simplistic mathematical correlation does not accurately conform to reality, but at least the number of 5 million is set as a reference point, especially for comparison among different systems.

Another stress-based fatigue study, which has been widely accepted, is the staircase method. While it is certainly a more efficient method [21,22,23,24,25,26,27,28,29], it remains less than optimal as it requires a relatively high number of specimens, which are sequentially tested to determine a mean value and, when applicable, the statistical distribution of the fatigue limit (see Figure 2).

The latter represents the load range below which there would be no failure for a particular number of cycles, previously defined as the reference. In this method, if a specimen fails before having reached the reference number of cycles of the fatigue limit definition, the next specimen is tested at an inferior load level, applying a determined decrement to the load range. If, instead, the specimen survives the cut-off number of cycles, the next specimen is tested at a superior load level, implementing a load increment equal to the previous decrement. Thus, each test is dependent on the previous one, with regard to the increment or decrement in the load, according to a prefixed range or step. This step can be defined based on a multiple or fraction of the standard deviation, which cannot be easily evaluated in advance in any case. In the staircase method graphics, the ordinate axis represents the load in Newtons, and the abscissa axis represents the sequentially ordered specimens. Failed specimens are generally represented by a cross, while the survivors, also called run-outs, are represented by a circle [9]. Assuming that there are no previous tests, approximately 20 specimens would be needed to estimate a 50% failure probability, 40 for 90% and 50 for 99% [21]. Data reduction techniques are applied to determine the statistical distribution of the results, such as the Dixon method or the Zhang and Kececioglu method [9].

Therefore, the staircase method provides an acceptably reliable fatigue limit for a pre-established number of cycles. However, the fatigue limit is only provided for a predetermined number of cycles, making it impossible to extrapolate the data for a higher number of cycles. In fact, a new staircase experimental program must be performed from the very beginning if a new predetermined number of cycles is required. Other limitations of the staircase method are pointed out in [22]. The reliability of the staircase is heavily dependent on the stress range step selected for the specific experimental program. The fact that around half of the tested specimens must by definition be run-outs renders this method inefficient since run-outs provide almost no primary information for the fatigue assessment. In addition, surviving (run-out) specimens are not taken into account in the statistical calculations. This clearly manifests the method’s insufficiency, due to the unnecessary number of tests to be carried out, apart from being the longest lasting ones. This ultimately makes this methodology an unnecessary and costly extension of the test program.

Fatigue testing results show inherently random character, making it necessary to facilitate the stress needed for a specimen to fail at a determined number of cycles, as well as the likelihood of failure. In other words, in order to guarantee a particular number of cycles until fracture, the corresponding stress range value will depend on the accepted probability of occurrence level. Given how widely dispersed the results usually are, a robust statistical data treatment is essential.

In dentistry, recent systematic reviews show that fatigue testing is usually programmed according to the S-N curves and, less often, according to the staircase method [30,31]. However, the aim of this paper is to justify the convenience of using more advanced probabilistic regression models to determine the S-N-P curve based on the extreme value theory—Weibull’s, in particular—as will be discussed below.

## 3. Previous and Current ISO Standard: Comparative Analysis

The International Organization for Standardization (ISO) 14801:2016 [32] is the international reference normative which establishes the parameters to carry out fatigue testing of endosseous dental implants, including their transmucosal abutment and premanufactured prosthetic components. This standardization allows the results to be comparable to one another.

Specifically, it describes the test method to be followed, including test machine characteristics, testing geometry, sample holder, load application, test environment, load frequency and waveform and test procedure characteristics.

Most of the outlined conditions are relatively easy to reproduce. However, the testing procedure used to obtain the fatigue limit proves to be highly problematic. The most recent version of the normative proposes two alternative procedures, which correspond to the two above-mentioned methods.

The first procedure consists of developing an S-N curve with an appropriate initial load of 80% of the failure load obtained from a previous static testing of similar geometry. In this case, it recommends testing at least two, and preferably three, specimens per load level. Thus, a minimum of four different range levels is established, in addition to a fifth inferior one, in which at least three specimens must survive the previously specified number of cycles (for example, 2 × 10^6^ or 5 × 10^6^ cycles). This value would represent the fatigue limit for this number of cycles.

Strict compliance with these conditions would therefore require using anywhere between 11 and 15 specimens per studied variable. Another inherent limitation of this procedure lies in the fact that it does not represent a reliable criterion, as it does not yield any reference in terms of probability in the determination of the fatigue limit, which, according to the normative, would require the survival of three specimens. Consequently, it is possible that either an excessively conservative or, by contrast, unsafe fatigue limit could potentially be defined for this number of cycles. If the proposed fatigue limit were associated with a 50% failure probability, which is in itself an unacceptable failure level, a favorable result (i.e., three run-outs when testing three specimens) would be achieved in 12.5% of cases, which represents an absolutely unacceptable risk if this fatigue limit were assumed as a reliable one. On the contrary, if the fatigue limit were associated with a 5% failure probability, which is relatively low but still represents an excessive failure percentage, the probability of a favorable result (again, three run-outs when testing three specimens) would be this time 86%. While much higher than before, it nevertheless still implies an unacceptable level of risk.

The 2016 version of the ISO 14,801 standard suggests an alternative method, the staircase method. In this case, the initial proposed load is equivalent to 50% of the static load resistance obtained from a similarly designed test (or, in its absence, using pre-existent data from comparable systems).

The step is defined as 10% of the maximum estimated load—that is, 5% of the failure load obtained from the static testing. Thus, the test begins at the maximum estimated load until the specimen failure or survival over 5 × 10^6^ cycles at a >2 Hz frequency. If the specimen fails, the subsequent specimen will be tested at a lower stress level, in a decrement equivalent to the predetermined step. If the specimen survives, the next one will be tested at an equally incremented stress level. The procedure must continue until at least four run-outs and four survivors are observed. Finally, the mean value of the maximum stated load, the standard deviation and the survival probabilities at 10, 50 and 90%, respectively, are calculated.

This method clearly represents an improvement over to the previous one, as it sets a fatigue limit for a determined number of cycles, which makes it possible to obtain the maximum stated load for this particular number of cycles (5 × 10^6^ cycles) more accurately and express it in terms of probability. Moreover, the first load level is lower, 50% of the maximum stated load in a static testing, rather than the 80% of the previous method. (Generally, this load turns out to be excessively high and does not provide relevant information to build the load cycles diagram).

However, the number of tested specimens in this methodology remains too low to guarantee a reliable fatigue limit. Again, as previously mentioned, it is important to note that the staircase method does not allow for the extrapolation of fatigue limit data for any number of cycles apart from what has been predetermined (be it 5, 10, 100 million or an infinite number of cycles). In fact, the new version of the ISO 14,801 standard does not even reference a “fatigue limit” at all, because it cannot effectively be obtained. Instead, it only establishes the “maximum load” of the previously defined arbitrary value of 5 × 10^6^ or 2 × 10^6^ cycles.

Similarly, the number of specimens necessary for guaranteeing a sufficiently reliable level is not specified a priori. The very nature of the methodology of the staircase method is unsatisfactory, since, by design, half of the specimens are survivors and therefore do not provide relevant information for the S-N curve definition. Clearly, this results in a considerable waste of both time and resources. Although experience tends to confirm that around 11 samples are usually needed to obtain a better estimation of the S-N curve, the problem formulated in these terms is considered unsolvable. Thus, the reader is referred to the proposed alternative methodology in the ProFatigue Software section.

In short, while the current ISO 14,801 standard offers improvements in the dynamic loading testing for dental implants, it nonetheless continues to suffer serious limitations in its effective application in the fatigue limit determination. The fact that most of the dental studies in this field do not satisfy the high number of specimens demanded by the normative only serves to confirm these inherent flaws [20,33,34,35,36,37,38].

## 4. ProFatigue Software

In opposition to the limited reliability of the aforementioned ISO 14,801 recommendations for the determination of the fatigue limit, the use of the ProFatigue software is proposed [39,40]. The program was developed by the IEMES (Structural Integrity: Materials and Structures) Research Group at the University of Oviedo, in collaboration with Prof. Castillo of the University of Cantabria and Empa-Dübendorf (Swiss Federal Laboratories for Materials Science and Technology). ProFatigue is based on the Weibull’s regression model proposed by Castillo-Canteli [41] in order to satisfy the physical and statistical conditions required by any valid fatigue model. It has been validated through its successful application to a wide range of materials and practical cases since its launch in 2014, including the fatigue assessment of bio-mechanical materials [42]. Its application in fatigue testing research in the field of dentistry would constitute a significantly improved alternative to those currently proposed in the ISO 14,801 standard (the simplified S-N curve definition and the staircase method), not only because of its accuracy in the fatigue limit estimation itself, but also its ability to provide the probabilistic definition of the whole S-N field of the studied implant or prosthetic component, which would result in much more precise predictions of the lifetime of these components [40].

The ProFatigue software represents a statistical tool that facilitates the probabilistic definition both of the S-N field (stress-number of cycles) and ε- N field (strain-number of cycles), according to the testing reference parameter. Consequently, it allows for the graphical representation of the predicted number of cycles needed to reach failure in a particular fatigue test from some experimental results, previously obtained and deduced from the applied load or deformation. To do so, the software estimates the five regression model parameters, three of which define the Weibull’s subjacent three-parameter distribution function.

The main advantages of this model include:The possibility of an optimized test planning campaign, which results in a reduction in the total number of tests required and its rationalization, by avoiding the undesirable outcome of run-outs, or at least reducing them to an acceptable number. Thus, the S-N field is determined by a number of tests not previously stated, and in any case, far fewer than those currently required by the ISO standard for the same level of confidence. Furthermore, given the normalization property that sustains the model, it is unnecessary to repeat the tests under the same load range, with any negative consequences for the completion of the statistical analysis (see Part 2). This contrasts with the alternative procedures to determine the S-N field presented by the current ISO standard, which require numerous tests for the same load range.The complete S-N field is reduced to a simple distribution function of the normalized variable, V, which is deduced as the combined expression of both stress range and number of cycles, through the expression *V = log(N/N_0_) log(Δσ/Δσ_0_)*, where *N_0_* is the number of limit cycles and *Δσ_0_* is the endurance limit for a theoretically infinite fatigue limit. This normalization allows the optimized planning of the testing campaign, in addition to the combined evaluation of different levels of test results as belonging to a single homogeneous group. This increases the reliability of the deduced model, and ultimately, it allows for the detection of possible bias in the joint distribution, thanks to the non-randomness of the results obtained from the different tested load levels, which would point out an anomalous behavior, that may be worthy of further study.The possibility of a simple calculation of the number of cycles for a given particular stress range and probability, or conversely, of the stress range for a predetermined number of cycles and probability, is a direct consequence of the S-N field analytic expression. This paragraph is directly related to paragraph 1, regarding its application as a means of support for fatigue program planning.

It is important to note that the fatigue programs related to the S-N field evaluation, including ProFatigue and other current programs, are focused on predicting the probability of reaching the terminal state of the implant, not necessarily failure. On the contrary, crack growth evolution, as a cumulative damage process inherent to the fatigue phenomenon, requires performing tests using sophisticated experimental techniques (such as potential drop, acoustic emission) to measure the physical crack growth evolution. In this case, a damage tolerance concept would be convenient in order to aid in the decision to either remove the implant or perform a restorative intervention. This is an interesting question to be considered for future implant research.

The program enables the visualization of the S-N field as the project progresses with the incorporation of new data (see Figure 3). In this way, a rational definition based on a probabilistic approach of the stress to be applied in the next test is established, and therefore the S-N field is defined according to an optimized strategy. This makes it possible to rationally save tested specimens and simultaneously achieve greater reliability in the parameter determination of the model [40].

Three run-out tests, carried out at different load ranges, could potentially be enough to secure an initial approach to the S-N field, although it would not guarantee a representative estimation of the scatter. Therefore, in order to obtain a reliable definition of the S-N field, 8–10 specimens would be sufficient in most cases, as long as there is an appropriate distribution of the load ranges (see Figure 4). A second level of analysis of the confident intervals would require applying either a “boot-strap” statistical method or the use of Bayesian techniques [43].

As is evidenced above, the ProFatigue software provides more comprehensive information and requires far fewer tests than the other methods outlined in this paper. The program offers an analytical definition of the S-N probabilistic field through percentiles associated with the desired probability and, in doing so, allows for the prediction of the lifetimes over the testing limits by extrapolation. This means that the fatigue limit for any probability can be calculated, not only for the 5 × 10^6^ or 2 × 10^6^ cycles predetermined by the ISO standard, but also for an infinite number of cycles (endurance limit or real fatigue limit), or any other desired number of cycles. Obviously, even the longest lasting implants will not exceed 80 × 10^8^ cycles (around 80 years in case they are placed in a young person), and therefore the calculation of the fatigue limit for an infinite number of cycles may not be of interest from a clinical point of view. However, in terms of in vitro tests, the ProFatigue software allows a much stricter and far more precise comparative analysis of specimens and a far more accurate real fatigue limit definition for each studied model.

The ProFatigue software is available free of cost through the link https://meteo.unican.es/temp/castie/Profatigue.html [39]. A user guide is also available, providing a detailed step-by-step overview of the program and installation.

Once downloaded, the program can be run in either manual or guided mode; however, only the guided mode enables the user to introduce the fatigue testing results step by step. Upon entering the experimental outcomes (number of cycles until failure, request range and identification of the specimens designated as survivors or run-outs, if any), the program supplies the estimated values of the five model parameters and displays the S-N field with the percentile curves corresponding to the chosen failure probability. By default, probabilities of 0, 1, 5, 50, 95 and 99% are shown. In addition, the implemented model in ProFatigue allows for the possibility of virtually transferring the run-outs to the position where the failure could be probabilistically expected. The program also permits a number of options in the estimation process, such as excluding any data of doubtful reliability or survivors tested below the fatigue limit. It is also possible to consider the scale effect, when appropriate to apply.

While the program itself is extremely straightforward, expert engineer support (particularly during the design of the testing strategy and the final global evaluation) is advisable in order to ensure a correct understanding of the criteria that rule the successive decision-making process in the test campaign. This support facilitates a real-time evaluation of the S-N field from the available results, with the objective of maintaining an optimized approach.

The ProFatigue program allows a probabilistically advanced estimation of the remaining lifetime of the implant but does not provide an estimation of the crack size evolution. The same would happen with any other current S-N evaluation software. To provide the progress of the crack size, fatigue tests must be performed on the implant material and crack growth must be registered by using sophisticate experimental techniques, such as potential drop, acoustic emission [44,45]. With this information, a damage tolerance concept could be applied to decide on possible withdrawal of the implant or the restorative intervention. This is an objective to be envisaged for a new investigation on implants.

## 5. Conclusions

The current ISO standard suffers from a number of limitations due to the inadequate selection of the probabilistic model recommended in the planning of the experimental program and the insufficient evaluation of the results obtained. This implies higher costs because of the large number of specimens tested and the long test duration, which should be avoided as much as possible.ProFatigue’s advanced statistical model optimizes these factors, making it an innovative and valuable tool that could be adapted for use in fatigue programs within the field of dentistry implant research.Correct implementation of the ProFatigue software makes it possible to improve the test programming procedure by reducing the number of tests and the total duration of the experimental design, while simultaneously guaranteeing reliability in the material characterization. Its efficacy and efficiency are supported by its long-standing use when applied to a wide range of experimental fatigue programs for different materials.

## Figures and Tables

**Figure 1 biology-09-00372-f001:**
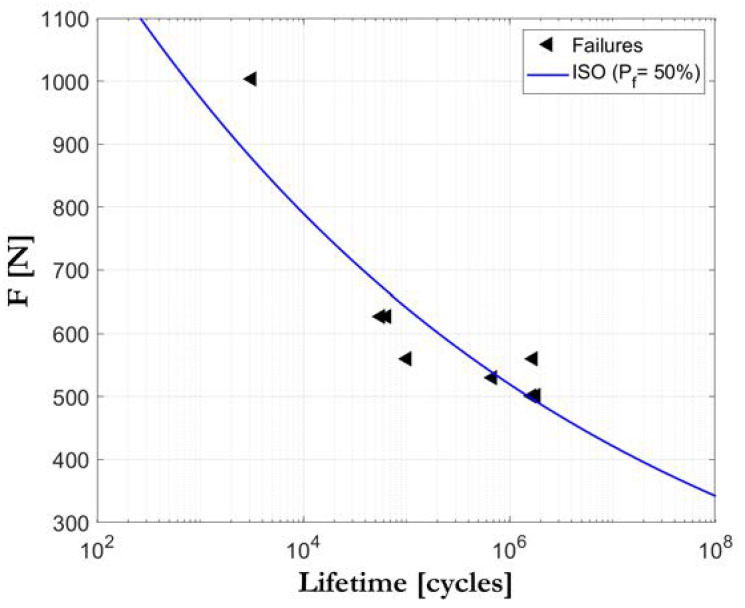
An example of an S-N field diagram. (Triangles: failed specimens).

**Figure 2 biology-09-00372-f002:**
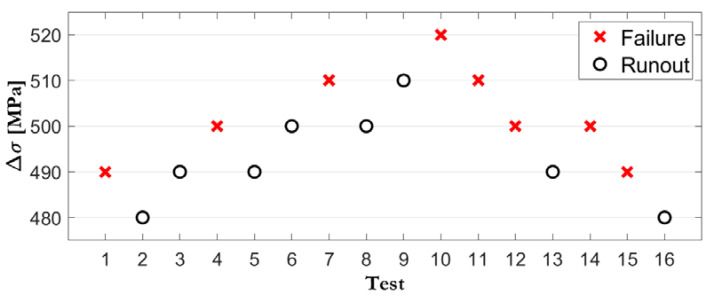
Example of sequential test results, subjected to the applied load for pre-established number of cycles to failure, in accordance with the staircase method. (Failed specimens in red; survivors (run-outs) in black).

**Figure 3 biology-09-00372-f003:**
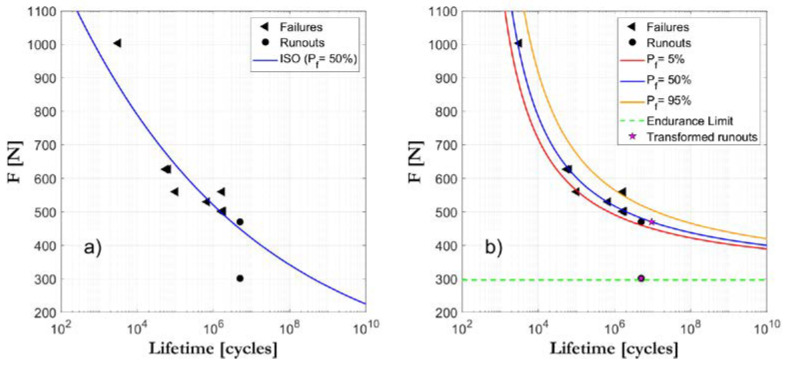
Comparative example between the ISO standard S-N field definition (**a**) and the ProFatigue S-N field definition (**b**) for the same studied case.

**Figure 4 biology-09-00372-f004:**
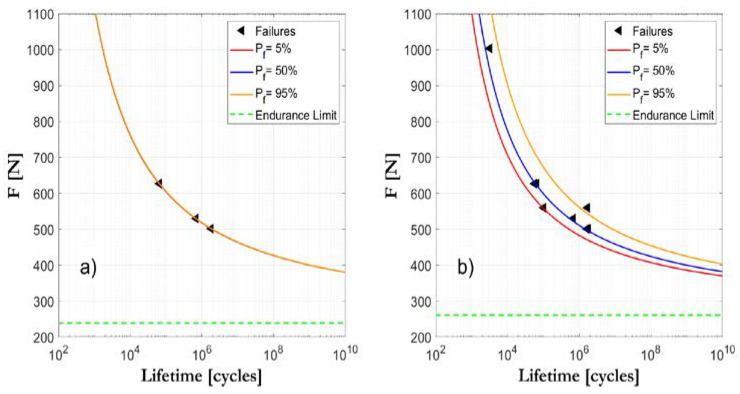
Example of assessment of the S-N curves by applying the ProFatigue program to the experimental data: provisional initial evaluation after the first three tested specimens (**a**); final evaluation, after eight tested specimens (**b**).

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
