# Peer review of "Optimized Planning and Evaluation of Dental Implant Fatigue Testing: A Specific Software Application"

_biology, 2020, doi:10.3390/biology9110372_

Round 1

Reviewer 1 Report

Comments from reviewer 2 and authors’s answears:

We thank the reviewer for the critical and constructive comments, which help the authors to improve the manuscript.

1. Question: “Is the work scientifically sound and not misleading? Lowest mark.

Answer

Concerning reviewer’s low mark about the form question “Is the work scientifically sound and not misleading?” we are not able to judge if the work is “misleading” but in what concerns “scientifically sound”, we would like to argue that the fatigue model proposed is summarized in the book [1], with more than 221 citations according to Google Scholar. The model is based on very demanding physical and statistical conditions (compatibility, limiting and asymptotic conditions and extreme value considerations [2]) which lead to a functional equation [3] (a very advanced field in applied Mathematics) the solution of which requires a very skilled knowledge in modeling. Accordingly, we hope the paper deserves to be considered “scientific sound”.

[1] Castillo E., Fernández-Canteli A., A unified statistical methodology for modeling fatigue damage, Springer 2009.

[2] Castillo E., Hadi A.S., Balakrishnan N., Sarabia J.M., Extreme value and related models with applications in engineering and science. New York: Wiley, 2005.

[3] Castillo E., Iglesias A., Ruíz-Cobo R.: Functional Equations in Applied Sciences. Elsevier B. V., Amsterdam (2005).

Perhaps the first evaluation was too rigorous. What I mean can be understood reading the answer to question number 5. In my opinion there are several conclusions that are too categorical and give to the paper a more biased aspect. Probably it’s derived from the passion and the perfect knowing of the software by the authors. Nevertheless, I think that a more moderate writing would provide a more impartial aspect.  

2. Question about “Abstract: In first phrase you talk about dental implants. Are you referring to implant-supported fixed dental prosthesis, aren’t you? Anyway, in line 28 you conclude that “helping researchers to gain a better understanding of the behaviour of dental materials”. It isn´t clear if this tool is useful to implant FDPs or all dental materials.”

Answer

The authors are indeed referring specifically to implant-supported fixed dental prosthesis, while it is true that both fixed and removable rehabilitations are susceptible of mechanical complications. Although this program may be used to optimize the mechanical characterization of different dental materials, this paper focuses on dental implant fatigue testing. In accordance with the reviewer’s suggestions, these aspects have been clarified in the new version of the abstract.

Much more clear. 

3. Question about “Introduction and general consideration: There are several affirmations that require be justified by previous research: for example: line 129 “is an efficient but not optimal method”. It sounds like a personal opinion.”

Answer

The authors already pointed out the limitations of the staircase method in [4]

The objective reasons for this comment are summarized in [4], among which we emphasize that about half of the tested specimens must be run-outs, by definition, which turns this method inefficient since runouts provide practically no primary information for the fatigue assessment.

Further, the staircase method only provides the fatigue limit for a predetermined limit number of cycles (to which the testing program is referred to), so that extrapolation for a higher number of cycles is not possible, i.e. a new staircase experimental program must be again performed from the very beginning for the new predetermined limit number of cycles. Finally, the reliability of the staircase is strongly dependent on the stress range step selected to be applied over the experimental program. Some additional comments are now included into the new version of the manuscript in accordance with Ref. [20].

[20] Castillo E., Ramos A., Koller R., López-Aenlle M., Fernández-Canteli A., A critical comparison of two models for assessment of fatigue data, Int. Journal of Fatigue, 30, 45-57, 2008.

The new comments added improve the explanation of this issue. From my point of view, a reference of this asseveration “While it is certainly a more efficient method” must be included in order to make it even clearer. The explanation is more than enough, but this sentence must be referenced.

4. Question about “Previous and current ISO Standard. Comparative analysis: The review and the considerations of these ISO recommendations are well developed and very useful for a critical review.”

Answer

We thank the reviewer for this positive comment.

You are welcome. I liked it very much. 

5. Question about “Software: It seems to be an interesting software that can be useful for these in nitro test fatigue. It’s well described and its advantages well explained too. Nevertheless, there aren’t any proof that proofs its advantages.”

Answer

The authors participated in a number of papers in which the positive application of the software is definitely proved. Nevertheless, since the present manuscript is not addressed to specialists, the authors believe that Ref. [31] provides a sufficient general description of the software program to show its potential interest.

[31] Fernández-Canteli A, Przybilla C, Nogal M, López Aenlle M, Castillo E. ProFatigue: A software program for probabilistic assessment of experimental fatigue data sets. Procedia Engineering

2014; 74:236–41.

We wonder why the reviewer considers that the second and third conclusions are not sufficiently justified. The authors participated since 1982 in a high number of international publications, research projects and industrial projects related to fatigue modeling both on theoretical and experimental aspects, particularly in a specific book of Springer (see Ref. [23]), (32?) specifically devoted to justify mathematically and statistically the background of the fatigue model used in the ProFatigue program.

We counted on the continuous collaboration since 1982 with Prof. Enrique Castillo, an international authority in the domain of extreme value statistics (see attached references):

- Castillo, E.; Extreme value theory in Engineering, Elsevier, 1980.

- Castillo E., Hadi A.S., Balakrishnan N., Sarabia J.M., Extreme value and related models with applications in engineering and science, New York, Wiley, 2005.

- Castillo, E.; Iglesias, A.; Ruíz-Cobo, R., Functional equations in applied sciences, Elsevier

Science, 1999. 

who set the basis of the fatigue model and, consequently of the ProFatigue program as testified by our common publications. All the statements included in the Conclusions, were exposed and supported statistically by the authors under the supervision of Prof. Castillo in all the previous publications. In particular, the normalizing property of the Weibull model, on which the ProFatigue program is based, allows all the experimental results to be pooled in a single sample cumulative distribution function, irrespective of the stress range at which the test is performed. This is a powerful property that ensures a reduction of the number of tests necessary to achieve certain reliability in the experimental assessment compared with other models currently applied (as the staircase method or the Basquin model). The normalizing property of the model propitiates the adequate test strategy since there is no necessity of replicating tests at the same stress range while the test program may be performed on line. In fact, a provisional model evaluation can be envisaged already with 4 initial tests as a result of which the subsequent stress range can be decided according with the parameter estimation obtained.

Unfortunately, the paper extension is limited so that we cannot deepen into the justification of the second and third conclusions by arguments and citations. We think that the international collaboration and common publications of our research group with prestigious colleagues of Univ. Bochum (Germany), Univ. Darmstadt (Germany), Univ. of Parma (Italy), Univ. of Porto (Portugal), IPM Brno Czech Academy of Sciences, Research Center Kazan Russian Academy of Sciences, etc. and the citations from a number of authors of the international fatigue community support our comments about the potentiality and advantages provided by the ProFatigue program proving that they are not gratuitous. See also the application to the fatigue assessment of biomechanical materials, now added to the reference list:

[33] Pascual F.J., Przybilla C., Gracia-Villa L., Puértolas J.A., Fernández Canteli A., Probabilistic

assessment of fatigue initiation data on highly crosslinked ultra high molecular weight polyethylenes, Journal of the Mechanical Behavior of Biomedical Materials, 15,190-198, 2012.

First of all I’d like to remark that the know-how and capability of the authors is beyond doubt. We are only evaluating the manuscript. Under my point of view, the conclusions are too ambitious and perhaps with some little modifications in their writing would sound less categorical and will give the manuscript a more impartial meaning. 

 For example avoiding unnecessary terms like: decisively, easily or clearly. 

“Correct implementation of the ProFatigue software makes it possible to optimize the tests programming procedure by significantly reducing the number of tests and the total duration of the experimental design, while simultaneously guaranteeing increased reliability in the characterization of the material. Its efficacy and efficiency are clearly demonstrated by its long-standing use in a wide range of cases of general characterization, applied to different materials subjected to fatigue. “

versus

“Correct implementation of the ProFatigue software makes it possible to optimize the tests programming procedure by significantly reducing the number of tests and the total duration of the experimental design, while simultaneously guaranteeing increased reliability in the characterization of the material. Its efficacy and efficiency are clearly demonstrated  are supported by its long-standing use in a wide range of cases of general characterization, applied to different materials subjected to fatigue. 

Author Response

REVIEWER 1

Comments from Reviewer and Authors’ answers:

We thank the reviewer for the critical and constructive comments, which help the authors to improve the manuscript.

  1. Question: “Is the work scientifically sound and not misleading? Lowest mark.

Answer

Concerning reviewer’s low mark about the form question “Is the work scientifically sound and not misleading?” we are not able to judge if the work is “misleading” but in what concerns “scientifically sound”, we would like to argue that the fatigue model proposed is summarized in the book [1], with more than 221 citations according to Google Scholar. The model is based on very demanding physical and statistical conditions (compatibility, limiting and asymptotic conditions and extreme value considerations [2]) which lead to a functional equation [3] (a very advanced field in applied Mathematics) the solution of which requires a very skilled knowledge in modeling. Accordingly, we hope the paper deserves to be considered “scientific sound”.

[1] Castillo E., Fernández-Canteli A., A unified statistical methodology for modeling fatigue damage, Springer 2009.

[2] Castillo E., Hadi A.S., Balakrishnan N., Sarabia J.M., Extreme value and related models with applications in engineering and science. New York: Wiley, 2005.

[3] Castillo E., Iglesias A., Ruíz-Cobo R.: Functional Equations in Applied Sciences. Elsevier B. V., Amsterdam (2005).

Perhaps the first evaluation was too rigorous. What I mean can be understood reading the answer to question number 5. In my opinion there are several conclusions that are too categorical and give to the paper a more biased aspect. Probably it’s derived from the passion and the perfect knowing of the software by the authors. Nevertheless, I think that a more moderate writing would provide a more impartial aspect.  

Answer:
We thank the reviewer’s comment and we will correct the possibly immoderate statements of the authors in the corresponding paragraphs.

  1. Question about “Abstract: In first phrase you talk about dental implants. Are you referring to implant-supported fixed dental prosthesis, aren’t you? Anyway, in line 28 you conclude that “helping researchers to gain a better understanding of the behaviour of dental materials”. It isn´t clear if this tool is useful to implant FDPs or all dental materials.”

Answer

The authors are indeed referring specifically to implant-supported fixed dental prosthesis, while it is true that both fixed and removable rehabilitations are susceptible of mechanical complications. Although this program may be used to optimize the mechanical characterization of different dental materials, this paper focuses on dental implant fatigue testing. In accordance with the reviewer’s suggestions, these aspects have been clarified in the new version of the abstract.

Much more clear. 

Answer:
We appreciate the reviewer’s comment.

  1. Question about “Introduction and general consideration: There are several affirmations that require be justified by previous research: for example: line 129 “is an efficient but not optimal method”. It sounds like a personal opinion.”

Answer

The authors already pointed out the limitations of the staircase method in [4]

The objective reasons for this comment are summarized in [4], among which we emphasize that about half of the tested specimens must be run-outs, by definition, which turns this method inefficient since runouts provide practically no primary information for the fatigue assessment.

Further, the staircase method only provides the fatigue limit for a predetermined limit number of cycles (to which the testing program is referred to), so that extrapolation for a higher number of cycles is not possible, i.e. a new staircase experimental program must be again performed from the very beginning for the new predetermined limit number of cycles. Finally, the reliability of the staircase is strongly dependent on the stress range step selected to be applied over the experimental program. Some additional comments are now included into the new version of the manuscript in accordance with Ref. [20].

[20] Castillo E., Ramos A., Koller R., López-Aenlle M., Fernández-Canteli A., A critical comparison of two models for assessment of fatigue data, Int. Journal of Fatigue, 30, 45-57, 2008.

The new comments added improve the explanation of this issue. From my point of view, a reference of this asseveration “While it is certainly a more efficient method” must be included in order to make it even clearer. The explanation is more than enough, but this sentence must be referenced.

 Answer

The suggested comment is taken into account and several references of the staircase method are now included in the manuscript:

- Dixon, W.J.; Mood, A.M. A Method for Obtaining and Analyzing Sensitivity Data. Journal of the American Statistical Association1948, 43, 109–26.

- Dixon, W.J. The Up-and-Down Method for Small Samples. Journal of the American Statistical Association 1965 60:967–78, DOI: 10.2307/2283398.

- Little, R.E.; Jebe, E.H.; Statistical design of fatigue experiments, Applied Science Publishers ltd, London , 1975, ISBN: 0853345872.

- Bruce, RD. An up-and-down procedure for acute toxicity testing, Fundam Appl Tox1985, 5, 151–7, DOI: 10.1016/0272-0590(85)90059-4

- Choi, S.C. Interval estimation of the LD50 based on an up-and-down experiment. Biometrics1990, 46, 485–92, DOI: 10.2307/2531453.

- Dixon, W.J; Design and analysis of quantal dose–response experiments (with emphasis on staircase designs) Dixon Statistical Associates, 1991, Los Angeles (CA), USA.

- Lipnick, R.L.; Cotruvo, J.A.; Hill, R.N.; Bruce, R.D..; Stitzel, KA.; Walker, A.P.; et al. Comparison of the up-and-down, conventional LD50 and fixed dose acute toxicity procedures ,. Food Chem Toxicol1995, 33, 223–31, DOI: 10.1016/0278-6915(94)00136-c.

- Vagerö, M.; Sundberg, R. The distribution of the maximum likelihood estimator in up-and-down experiments for quantal dose–response data, J Biopharmaceut Stat1999, 9(3), 499–519,https://doi.org/10.1081/BIP-100101190.

Question about “Previous and current ISO Standard.

Comparative analysis: The review and the considerations of these ISO recommendations are well developed and very useful for a critical review.”

Answer

We thank the reviewer for this positive comment.

You are welcome. I liked it very much. 

 Answer:
We are very pleased.

  1. Question about “Software: It seems to be an interesting software that can be useful for these in nitro test fatigue. It’s well described and its advantages well explained too. Nevertheless, there aren’t any proof that proofs its advantages.”

Answer

The authors participated in a number of papers in which the positive application of the software is definitely proved. Nevertheless, since the present manuscript is not addressed to specialists, the authors believe that Ref. [31] provides a sufficient general description of the software program to show its potential interest.

[31] Fernández-Canteli A, Przybilla C, Nogal M, López Aenlle M, Castillo E. ProFatigue: A software program for probabilistic assessment of experimental fatigue data sets. Procedia Engineering

2014; 74:236–41.

We wonder why the reviewer considers that the second and third conclusions are not sufficiently justified. The authors participated since 1982 in a high number of international publications, research projects and industrial projects related to fatigue modeling both on theoretical and experimental aspects, particularly in a specific book of Springer (see Ref. [23]), (32?) specifically devoted to justify mathematically and statistically the background of the fatigue model used in the ProFatigue program.

We counted on the continuous collaboration since 1982 with Prof. Enrique Castillo, an international authority in the domain of extreme value statistics (see attached references):

- Castillo, E.; Extreme value theory in Engineering, Elsevier, 1980.

- Castillo E., Hadi A.S., Balakrishnan N., Sarabia J.M., Extreme value and related models with applications in engineering and science, New York, Wiley, 2005.

- Castillo, E.; Iglesias, A.; Ruíz-Cobo, R., Functional equations in applied sciences, Elsevier

Science, 1999. 

who set the basis of the fatigue model and, consequently of the ProFatigue program as testified by our common publications. All the statements included in the Conclusions, were exposed and supported statistically by the authors under the supervision of Prof. Castillo in all the previous publications. In particular, the normalizing property of the Weibull model, on which the ProFatigue program is based, allows all the experimental results to be pooled in a single sample cumulative distribution function, irrespective of the stress range at which the test is performed. This is a powerful property that ensures a reduction of the number of tests necessary to achieve certain reliability in the experimental assessment compared with other models currently applied (as the staircase method or the Basquin model). The normalizing property of the model propitiates the adequate test strategy since there is no necessity of replicating tests at the same stress range while the test program may be performed on line. In fact, a provisional model evaluation can be envisaged already with 4 initial tests as a result of which the subsequent stress range can be decided according with the parameter estimation obtained.

Unfortunately, the paper extension is limited so that we cannot deepen into the justification of the second and third conclusions by arguments and citations. We think that the international collaboration and common publications of our research group with prestigious colleagues of Univ. Bochum (Germany), Univ. Darmstadt (Germany), Univ. of Parma (Italy), Univ. of Porto (Portugal), IPM Brno Czech Academy of Sciences, Research Center Kazan Russian Academy of Sciences, etc. and the citations from a number of authors of the international fatigue community support our comments about the potentiality and advantages provided by the ProFatigue program proving that they are not gratuitous. See also the application to the fatigue assessment of biomechanical materials, now added to the reference list:

[33] Pascual F.J., Przybilla C., Gracia-Villa L., Puértolas J.A., Fernández Canteli A., Probabilistic

assessment of fatigue initiation data on highly crosslinked ultra high molecular weight polyethylenes, Journal of the Mechanical Behavior of Biomedical Materials, 15,190-198, 2012.

First of all I’d like to remark that the know-how and capability of the authors is beyond doubt. We are only evaluating the manuscript. Under my point of view, the conclusions are too ambitious and perhaps with some little modifications in their writing would sound less categorical and will give the manuscript a more impartial meaning. 

 For example avoiding unnecessary terms like: decisively, easily or clearly. 

  • ProFatigue’s advanced statistical models decisively optimize those factors, making it an innovative and valuable tool that could be easily adapted for use in fatigue programs within the field of dentistry implant research.

“Correct implementation of the ProFatigue software makes it possible to optimize the tests programming procedure by significantly reducing the number of tests and the total duration of the experimental design, while simultaneously guaranteeing increased reliability in the characterization of the material. Its efficacy and efficiency are clearlydemonstrated by its long-standing use in a wide range of cases of general characterization, applied to different materials subjected to fatigue. “

versus

“Correct implementation of the ProFatigue software makes it possible to optimize the tests programming procedure by significantly reducing the number of tests and the total duration of the experimental design, while simultaneously guaranteeing increased reliability in the characterization of the material. Its efficacy and efficiency are clearly demonstrated  are supported by its long-standing use in a wide range of cases of general characterization, applied to different materials subjected to fatigue. 

We thank the reviewer for the positive comments. The proposal is accepted and the initial text of the conclusions is subsequently corrected.

Reviewer 2 Report

In this paper entitled ”Optimized Planning and Evaluation of Dental Implant fatigue Testing: A Specific Software  Application.” the authors presented a review of the current ISO 14801 which provide guidelines for the evaluation of dental implant mechanical reliability and moreover, they intriduced a new software called ProFatigue that optimizes fatigue testing of implant-supported prostheses.

The abstract is well organized and clearly presents the topic under consideration with distinct references to the aim of the review, referring several times how the software examined evaluates implant-supported fixed dental prosthesis.

In the part regarding the general considerations on load, fatigue and SN Curves, in particular the experimental techniques to obtain information related to fatigue behavior, and also to predict lifetime of implant-restoration complex materials, which are SN Curves and the Staircase Method are well discussed. In the section concerning the current ISO 14801 Standard the authors described improvements in the dynamic loading testing for dental implants, and analyzed the important limitations in its effective application in the fatigue limit determination due to the high number of specimens demanded by the norm.

 In particular, the authors implemented the part regarding the limitations of the Starcase method highlighting how the reliability of the method depends on the stress range step selected for the specific experimental program, supporting what is expressed by bibliographical references.

In the last section, the authors described the ProFatigue software as a tool to optimize the tests' programming procedure by reducing the number of tests and the total duration of the experimental design as supported by previous studies.

The manuscript is well written and organized, it also takes into consideration a debated topic in the implant-prosthetic field and allows to have a good overview of the tests currently used to evaluate the fatigue from loading of prostheses on implants. English revision by a native speaker is required and the introduction section may be implemented by other reviews or other studies concerning the causes of implant-prosthetic failures. in addition, the limitations of the method described must be included and the last part concerning the physical crack growth evolution analysis with studies reported in the literature must be implemented

Author Response

REVIEWER 2

Comments from Reviewer and Authors’ answers:

In this paper entitled ”Optimized Planning and Evaluation of Dental Implant fatigue Testing: A Specific Software  Application.” the authors presented a review of the current ISO 14801 which provide guidelines for the evaluation of dental implant mechanical reliability and moreover, they intriduced a new software called ProFatigue that optimizes fatigue testing of implant-supported prostheses.

The abstract is well organized and clearly presents the topic under consideration with distinct references to the aim of the review, referring several times how the software examined evaluates implant-supported fixed dental prosthesis.

In the part regarding the general considerations on load, fatigue and SN Curves, in particular the experimental techniques to obtain information related to fatigue behavior, and also to predict lifetime of implant-restoration complex materials, which are SN Curves and the Staircase Method are well discussed. In the section concerning the current ISO 14801 Standard the authors described improvements in the dynamic loading testing for dental implants, and analyzed the important limitations in its effective application in the fatigue limit determination due to the high number of specimens demanded by the norm.

 In particular, the authors implemented the part regarding the limitations of the Starcase method highlighting how the reliability of the method depends on the stress range step selected for the specific experimental program, supporting what is expressed by bibliographical references.

In the last section, the authors described the ProFatigue software as a tool to optimize the tests' programming procedure by reducing the number of tests and the total duration of the experimental design as supported by previous studies.

The manuscript is well written and organized, it also takes into consideration a debated topic in the implant-prosthetic field and allows to have a good overview of the tests currently used to evaluate the fatigue from loading of prostheses on implants. English revision by a native speaker is required and the introduction section may be implemented by other reviews or other studies concerning the causes of implant-prosthetic failures. in addition, the limitations of the method described must be included and the last part concerning the physical crack growth evolution analysis with studies reported in the literature must be implemented

We thank the reviewer for the positive comments. The suggestion of adding some references about critical crack growth is taken into account, and the following references are added:

- Blasón, S.; Fernández-Canteli, A.; Rodríguez, C.; Castillo, E. Retroextrapolation of crack growth curves using phenomenological models based on cumulative distribution functions of the generalized extreme value family, Int. J. of Fatigue 2020, https://doi.org/10.1016/j.ijfatigue.2020.105897

- Kim, Y.J.; You, H.; Kim, S.J.; Yun G.J. Effects of porosity on the fatigue life of polyamide 12 considering crack initiation and propagation, Advances Composite Materials. 2020.

Regarding the question about the introduction section:

Although several factors leading to implant-prosthetic failures were cited and categorized as either patient dependent or professional dependent, following the reviewer recommendation, two systematic reviews with meta-analysis have been included to support the statement that complications may have a mechanical, biological or esthetic origin (Line 35-36). As the paper focuses on mechanical evaluation, specifically on fatigue tests, we hope that this implementation will be enough to satisfy the change required.

 - Hu, M.L.; Lin, H.; Zhang, Y.D.; Han, J.M. Comparison of technical, biological, and esthetic parameters of ceramic and metal-ceramic implant-supported fixed dental prostheses: A systematic review and meta-analysis. J. Prosthet. Dent. 2020, 124, 26-35.

- Gaddale, R.; Mishra, S.K.; Chowdhary, R. Complications of screw- and cement-retained implant-supported full-arch restorations: a systematic review and meta-analysis. Int. J. Oral Implantol. 2020, 13, 11-40.

Finally, we remark that the text has been already revised by a native English speaker.

Round 2

Reviewer 2 Report

The authors made some major changes, but some minor errors are still present. In fact there are still some formatting errors in the entire manuscript. For example, the images are of low quality, please insert higher quality images. Different fonts are used for the various images. I suggest referring to the journal guidelines. Furthermore, from line 316 to 322 there would seem to be a different formatting. It is also important to double-check the bibliography since there are several errors: n 11 12 13 21 27 32 41. Please double-check the entire text.

Author Response

Prof.Dr. Ángel Álvarez Arenal

University of Oviedo.

Servicio de Prótesis y Oclusión. Clínica Universitaria de Odontología (CLUO).

C/ Catedrático Serrano, s/n. C.P.: 33006, Oviedo.

[email protected]

Dr. Prof. Dr. Juan Carlos Prados-Frutos
Dr. María Prados-Privado

Guest Editors

Biology. Special Issue "New Trends in Bioengineering in Osseointegration and Dental Implants"

October 28, 2020.

Dear Dr. Prof. Dr. Juan Carlos Prados-Frutos and Dr. María Prados-Privado,

On behalf of Dr. Álvarez-Arenal, I would like to resubmit the original article entitled “Optimized planning and evaluation of dental implant fatigue testing: a specific software application”, to be further considered for publication as a review in Biology, particularly in the Special Issue "New Trends in Bioengineering in Osseointegration and Dental Implants".

We are very grateful for the reviewer’s suggestions and we hope that the subsequent changes are properly implemented. All changes have been highlighted (now in blue) and responses to the reviewer’s comments are also included below.

In addition, we have improved the figures quality and revised the format of all references (including ISBN references).

We believe that this manuscript may be now suitable for its publication in the Special Issue "New Trends in Bioengineering in Osseointegration and Dental Implants". It offers the application of sophisticated and easy to use software to optimize experimental research in mechanical studies that may help researchers to carry out dental implants fatigue testing more efficiently.

Thank you for your consideration!

Yours sincerely,

Ms. Marta García González

Doctoral Student. Department of Dental Prostheses and Occlusion.

University of Oviedo.

REVIEWER 2

Comments from Reviewer and Authors’ answers:

The authors made some major changes, but some minor errors are still present. In fact there are still some formatting errors in the entire manuscript. For example, the images are of low quality, please insert higher quality images. Different fonts are used for the various images. I suggest referring to the journal guidelines. Furthermore, from line 316 to 322 there would seem to be a different formatting. It is also important to double-check the bibliography since there are several errors: n 11 12 13 21 27 32 41. Please double-check the entire text.”

We appreciate the reviewer’s recommendations, which will improve the quality of this paper.

We have made important changes in the images and we hope that the result will be now more satisfactory. The fonts are now homogeneous and some minor corrections have been made in the descriptions to accommodate the new pictures.

Formatting discrepancies have been corrected in lines 316 to 322 (now 318 to 323), as well as in the Keyword section (lines 43 to 44). Other small changes in the format have also been performed in lines 309 to 314, 399, 427, 430, 444, 451, 453 and 489.

Concerning the bibliography, ISBN numbers have been included in the book references (11, 12, 13, 21, 24 and 41). Unfortunately, there is no ISBN number for reference 27, so we have cited it following the example of another journal from the same publisher:

Salga, M.S.; Ali, H.M.; Abdullah, M.A.; Abdelwahab, S.I.; Hussain, P.D.; Hadi, A.H.A. Mechanistic Studies of the Anti-Ulcerogenic Activity and Acute Toxicity Evaluation of Dichlorido-Copper(II)-4-(2-5-Bromo-benzylideneamino)ethyl) Piperazin-1-ium Phenolate Complex against Ethanol-Induced Gastric Injury in Rats. Molecules 201116, 8654-8669.

  1. Dixon, W.J. Design and Analysis of Quantal Dose-Response Experiments (with Emphasis on Staircase Designs); Dixon Statistical Associates: Los Angeles, CA, USA, 1991. [Google Scholar]

With regard with the reference 32, we have proceeded similarly. This time from another paper from the same publisher:

Rojo, R.; Prados-Privado, M.; Reinoso, A.J.; Prados-Frutos, J.C. Evaluation of Fatigue Behavior in Dental Implants from In Vitro Clinical Tests: A Systematic Review. Metals 20188, 313.

  1. Organization, I.S. ISO 14801: Dentistry—Implants—Dynamic Fatigue Test for Endosseous Dental Implants; ISO: Geneve, Switzerland, 2007. [Google Scholar]

This manuscript is a resubmission of an earlier submission. The following is a list of the peer review reports and author responses from that submission.

Round 1

Reviewer 1 Report

the manuscript shows an interesting software for mechanical analysis in implantology. Although the text is very well written, it does not fully adapt to the format of the journal. I think that the authors should raise it with the sections, which in a generic way, are raised in the guides for authors. Although the manuscript is interesting, I believe that it does not adapt to the format and type of articles published by the journal Biology, I would advise the authors and send it to another MdPI journal more adapted to the issue, such as Materials. I think it would be interesting to increase the number of citations of the article to publish it in a field of materials engineering

Reviewer 2 Report

Optimized Planning and Evaluation of Dental Implant fatigue Testing: A Specific Software Application.

Abtract: 

  • In first phrase you talk about dental implants. Are you referring to implant-supported fixed dental prosthesis, aren’t you? Anyway, in line 28 you conclude that “helping researchers to gain a better understanding of the behaviour of dental materials”. It isn´t clear if this tool is useful to implant FDPs or all dental materials. 

1,2- Introduction and general consideration

There are several affirmations that require be justified by previous research: for example: line 129 “is an efficient but not optimal method”. It sounds like a personal oppinion. 

3- Previous and Current ISO Standard. Comparative Analysis.

The review and the considerations of these ISO recommendations are well developed and very useful for a critical review. 

4- Software: It seems to be an interesting software that can be useful for these in nitro test fatigue. It’s well described and its advantages well explained too. Nevertheless, there aren’t any proof that proofs its advantages. 

5.- Conclusions: 

Although the first conclusion can be interesting and perfectly deduced by the explained throughout the manuscript, the second and third ones are weakly demostrsated. I think that it cannot  be concluded in a review manuscript like this. 

From my point of view, there are two parts in this manuscript. 

1.- The review and conclusion about ISO recommendations can be useful and is well conducted.

2.- The specific software seems to be promising for a theoretical point of view but hadn’t been applied in a previous or preliminary studies. I recommended to the authors to make in vitro comparative studies with both systems to demonstrate that the tool presented increases the reliability of the material characterization as affirmed in the conclusions.    

Reviewer 3 Report

In this manuscript entitled :” Optimized Planning and Evaluation of Dental Implant fatigue Testing: A Specific Software  Application.” the authors presented a review of the current ISO 14801 which provide guidelines for the evaluation of dental implant mechanical reliability and in addition, the authors presented a new software named ProFatigue that optimizes fatigue testing of implant-supported prostheses. The abstract is well organized and clearly presents the topic under consideration with distinct references to the aim of the review.

In the part concerning the general considerations on load, fatigue and SN Curves, in particular the experimental techniques to obtain information related to fatigue behavior, and also to predict lifetime of implant-restoration complex materials, which are SN Curves and the Staircase Method are well discussed. In the section concerning the current ISO 14801 Standard the authors described improvements in the dynamic loading testing for dental implants, and analyzed the important limitations in its effective application in the fatigue limit determination due to the high number of specimens demanded by the norm.
In the last section, the authors decribed the ProFatigue software as a tool to optimize the tests' programming procedure by diminishing the number of tests and reducing the total duration of the experimental design as supported by previous studies.
The paper is well written and organized, it also takes into consideration a debated topic in the implant-prosthetic field and allows to have a good overview of the tests currently used to evaluate the fatigue from loading of prostheses on implants. English revision by a native speaker is required and the introduction section may be implemented by other reviews or other studies concerning the causes of implant-prosthetic failures.

Reviewer 4 Report

This article analyzes the ISO 14801 standard and suggests the use of software to perform the fatigue analysis of dental implants. However, as it is a literature review, more data should be collected and described so that readers understand the importance of the proposed and discussed program.

The article provides explanations about the advantages of the software, as a solution to the implant fatigue problem, however, the physical tests, in my opinion, should continue to exist because they bring important information that makes it possible to characterize the implants and continue improving these products. I would like the authors to have a more in-depth discussion on this point.

Another point that should be discussed is the weakness of the software in relation to the corrections of the flaws presented by the implants after the fatigue analysis.